# The Effect of Matrix Stiffness on Human Hepatocyte Migration and Function—An In Vitro Research

**DOI:** 10.3390/polym12091903

**Published:** 2020-08-24

**Authors:** Tingting Xia, Runze Zhao, Fan Feng, Li Yang

**Affiliations:** Key Laboratory of Biorheological Science and Ministry of Education, Bioengineering College, Chongqing University, Chongqing 400044, China; zrz23@cqu.edu.cn (R.Z.); ffan2016@cqu.edu.cn (F.F.)

**Keywords:** substrate stiffness, hepatocyte, migration, hepatocyte migration, hepatocyte function

## Abstract

The extracellular matrix (ECM) regulates cellular function through the dynamic biomechanical and biochemical interplay between the resident cells and their microenvironment. Pathologically stiff ECM promotes phenotype changes in hepatocytes during liver fibrosis. To investigate the effect of ECM stiffness on hepatocyte migration and function, we designed an easy fabricated polyvinyl alcohol (PVA) hydrogel in which stiffness can be controlled by changing the concentration of glutaraldehyde. Three stiffnesses of hydrogels corresponding to the health of liver tissue, early stage, and end stage of fibrosis were selected. These were 4.8 kPa (soft), 21 kPa (moderate), and 45 kPa (stiff). For hepatocytes attachment, the hydrogel was coated with fibronectin. To evaluate the optimal concentration of fibronectin, hydrogel was coated with 0.1 mg/mL, 0.01 mg/mL, 0.005 mg/mL, or 0.003 mg/mL fibronectin, and the migratory behavior of single hepatocyte cultured on different concentrations of fibronectin was analyzed. To further explore the effect of **s**ubstrate stiffness on hepatocyte migration, we used a stiffness controllable commercial 3D collagen gel, which has similar substrate stiffness to that of PVA hydrogel. Our result confirmed the PVA hydrogel biocompatibility with high hepatocytes survival. Fibronectin (0.01 mg/mL) promoted optimal migratory behavior for single hepatocytes. However, for confluent hepatocytes, a stiff substrate promoted hepatocellular migration compared with the soft and moderate groups via enhancing the formation of actin- and tubulin-rich structures. The gene expression analysis and protein expression analysis showed that the stiff substrate altered the phenotype of hepatocytes and induced apoptosis. Hepatocytes in stiff 3D hydrogel showed a higher proportion of cell death and expression of filopodia.

## 1. Introduction

Liver fibrosis is a common problem in patients with a long-standing history of iterative liver injury [1]. It is characterized by aberrant extracellular matrix (ECM) deposition and leads to increased tissue stiffness. [2]. As fibrosis develops, hepatocyte regeneration and excess ECM deposition further distorts the intrahepatic structures and inhibits septal angiogenesis. This eventually results in a progressive loss of hepatic function. Clinical diagnosis commonly uses transient elastography (TE) to assess liver stiffness. Conclusions from the clinical data indicated that the stiffness of healthy liver tissue is 1.5–4.5 kPa, while stiffness during the early and end stages of fibrosis were 4.1–21.9 and 16.3–48 kPa, respectively [3,4].

Normally the ECM in the liver is composed of various types of collagen, including type I, III, IV, V, and VI, but are gradually replaced by collagens I and III during fibrosis. Resident cells interact with the ECM to dynamically maintain tissue homeostasis and integrity. Thus, ECM stiffness is a key mechanical effector of local cellular behavior. Studies have indicated that increased ECM stiffness in chronic fibrotic diseases could affect the migratory behavior of resident cells [5,6]. It is well-known that local cells respond to mechanical stimuli induced through the ECM via their cytoskeletal structures [7]. Rigid ECM promotes morphological changes in resident cells, such as the assembly of lamellipodia and lamellae, which are specialized actin-rich structures that facilitate cellular migration along the ECM [8]. Since Pelham and Wang demonstrated that a stiffer polyacrylamide (PA) substrate promoted cell migration [9], there have been several substrate stiffness and cellular migration related studies. For instance, human skin fibroblasts grow on polyethylene glycol (PEG) gel, and, when the stiffness of PEG gel is from 95 Pa to 4.3 kPa, the cell migration speed decreases from 0.81 μm/min to 0.38 μm/min [10]. As one of the features of liver fibrosis, the stiffness of liver tissue is gradually increased during the progress of fibrosis. However, very few studies focus on the effect of matrix stiffness on the migratory behavior of hepatocytes during liver fibrosis.

Hepatocytes are the main parenchymal cells and functional units of the liver. Liver fibrosis is a complex process, which affects hepatocyte function and will trigger a series of events [11,12,13]. Previous studies have shown that hepatocytes contributed to the process of liver fibrosis through changing migratory behavior and biological functions. Hepatocytes can undergo epithelial-mesenchymal phenotypic transition, which increase the migratory ability and changes in the mechanical properties of hepatocytes can regulate cell migration and mechanical signal transduction [14,15,16]. However, at present, whether matrix stiffness, a factor that significantly changes in the process of liver fibrosis, can regulate hepatocytes participate in the process of liver fibrosis is unknown. Previous study has demonstrated that hepatocytes cultured on soft heparin gels were secreting 5-fold higher levels of albumin compared to cells cultured on stiff heparin [17]. However, research concerning the effects of pathological ECM stiffness on hepatocytes is still underdeveloped.

Therefore, we established a stiffness-controllable and tunable polyvinyl alcohol (PVA) hydrogel system to mimic the changes in liver tissue stiffness during the various stages of fibrogenesis. The migratory behavior of hepatocytes and changes to cytoskeletal proteins on substrates of different stiffness was investigated. The effect of ECM ligands on cellular migration was also examined. Gene and protein expression were measured to investigate the effect of substrate stiffness on hepatocellular functions. Additionally, the migrator behavior of hepatocytes in 3D substrate was also analyzed. Our results revealed that an increase in substrate stiffness promoted the migration of hepatocytes by enhancing formation of actin- and tubulin-rich structures. Moreover, stiff substrates altered the phenotype of hepatocytes and induced apoptosis.

## 2. Materials and Methods

### 2.1. PVA Hydrogel Preparation

PVA hydrogel was prepared as our previous study [18]. Briefly, 8% (*w/v*) PVA solution was made; for example, 32 g PVA powder (Aladdin, Los Angeles, CA, USA) was dissolved into 400 mL distilled water at 90 °C. PVA solution was polymerized after adding 25% (*m/v*) glutaraldehyde (Ourchem, Shanghai, China) and hydrochloric acid at room temperature. The stiffness of PVA hydrogel was determined by the concentration of glutaraldehyde. Before cell seeding, the hydrogels were incubated with fresh 200 mmol/L *N*′-a-hydroxythylpiperazine-*N*′- ethanesulfanic acid, HEPES (H3375, Sigma–Aldrich, St. Louis, MO, USA) at pH 7.4 for 24 h and were washed by sterilized HEPES three times. Then, the hydrogels were soaked in 75% ethanol and treated with ultraviolet light more than 30 min for sterilization. Next, the PVA hydrogels were totally covered with 1 mM sulfo-SANPAH (22,598, Thermo Scientific, Rockford, IL, USA) solution and irradiated for 10 min using an ultraviolet lamp. Then, the operation was repeated. The hydrogels were washed by HEPES three times to remove residual cross linker. After that, 10 μg/mL fibronectin solution was coated to the PVA hydrogels and reacted for 12 h at 4 °C. Fibronectin-coated PVA hydrogels were washed twice with phosphate buffer solution (PBS) to remove unreacted fibronectin before planting cells.

### 2.2. PVA Hydrogel Young’s Modulus Analysis

The Young’s modulus of PVA hydrogels with different concentration of glutaraldehyde were measured by atomic force microscopy (AFM, JPK Instruments, NanoWizard II, Germany). The silicon nitride spherical probe used in the experiment was purchased from Nanosensors (Neuchatel, Switzerland) with a diameter of 4 μm, a spring constant of 0.2 N/m, and a poisson’s ratio of 0.5. The selected tapping mode was used to detect Young’s modulus. Force-displacement curves were used to calculate the Young’s modulus of hydrogel.

### 2.3. Cell Culture on the PVA Hydrogel

The immortalized human hepatocytes cells line L-02 was purchased from cell library of the Chinese Academy of Sciences (Catalog number: GNHu 6). Cells were seeded in RPMI 1640 (Roswell Park Memorial Institute, Buffalo, NY, USA) medium containing 10% fetal bovine serum, 100 unit/mL penicillin, and streptomycin at 37 °C, 5% CO_2_. The medium was changed every 2 days, and cells were passaged at 80% confluence using 0.25% trypsin-EDTA (ethylene diamine tetraacetie acid) solution. The cells were seeded at the density of 20,000 cells/ cm^2^ onto the hydrogel cast in 60-mm culture dishes or 6-well plates.

### 2.4. Cell Culture in the 3D Hydrogel

A stiffness controllable 3D Col-Tgel (collagen and gel) (Weihui Biotechnology, Beijing, China) was used for analyzing the migrator behavior of hepatocytes in 3D substrate stiffness. The Collagen-I (Col) solution was preheated at 55 °C for about 5 min until the collagen gel melted into a liquid state, and then it was cooled down at room temperature for 5 min before proceeding. The transglutaminase (TG) powder was taken out from −80 °C and warmed at room temperature; it was then dissolved in 100 μL sterile deionized water according to the instructions. After removing the medium, the digested hepatocytes and the Col solution were mixed by gently pipetting to avoid air bubbles. The TG solution was added according to the proportion and mixed gently by pipetting. In this study, 20 μL of 3D Col-Tgel mixture was seeded with 2 × 10^3^ hepatocytes. 20 μL of the 3D Col-Tgel-cell mixture were placed dropwise into the 24-well plate cell special circular slide (diameter 14 mm), and incubated in a cell incubator at 37 °C for 45 min. After the gel solidified, it was placed into the 24-well plate; then 1 mL of 10% FBS medium was added into the plate to ensure that the medium completely covered the gel, and the cells were cultured for subsequent testing.

### 2.5. PVA Hydrogel Biocompatibility Analysis

Cell Live/Dead kit were used to detect cellular apoptosis on PVA hydrogel and 3D hydrogel with following protocol: First, Calcein-acetoxymethylester (Calein-AM) and propidium iodide (PI) stock resolutions were diluted with 1 × assay buffer for preparing working solutions. Second, the complete medium was removed and then washed with 1 × assay buffer 2–3 times. Last, cells were incubated with 1ml Calcein-AM/PI staining solution at 37 °C for 15 min. Images were captured by using fluorescence microscopy (Olympus-IX71, Tokyo, Japan).

### 2.6. Time-Lapse Video Microscopy and Cell Migration Analysis

Time-lapse phase-contrast images of cell movement were captured every 5 min over a 6-h period using an inverted microscope with a 20× objective (LSM510 Meta, Zeiss, Oberkochen, Germany). Hydrogels were cast in 24-well plate and cells were seeded for 12 h before tracking. Images were collected and analyzed using Axio Vision Rel (version 4.8) software (Zeiss, Oberkochen, Germany). Image analysis was performed with Image J software (NIH, Bethesda, MD, USA).

### 2.7. Cell Migration Analysis in 3D Hydrogel

The assay was performed using 24-well transwell chambers containing polycarbonate filters with a pore size of 8 μm. The lower compartment was filled with 10% FBS medium. 1 × 10^4^ hepatocytes were seeded in 100 μL 3D gels on the upper compartment of the chamber. After incubation at 37 °C for 96 h, the cells and gels on the upper surface of the filter were removed by wiping. Cells that passed the filter were fixed with 4% paraformaldehyde for 20 min and stained with 0.2% crystal violet for 15 min. Images were captured by using phase contrast microscopy (Olympus-IX71, Tokyo, Japan).

### 2.8. Real-Time Quantitative Reverse Transcription PCR (RT-qPCR)

After 24-h culture on different stiffness of substrates and confluent growth cells were harvested. The total RNA was extracted using an Rneasy Mini Kit (TianGen, Beijing, China) following the manufacturer’s protocol. Total RNA was reverse-transcribed into cDNA by using a Prime Script RT reagent kit with gDAN Eraser. The cDNA products were then amplified using SYBR green PCR Premix (TaKaRa, Tokyo, Japan) and following specific primers. Real-time RT-PCR reactions were conducted in triplicate using CFX96 Real-Time PCR Detection System (Bio-Rad Laboratories, Hercules, CA, USA). Primer sequences were listed below: albumin, forward sequence (5′-3′): TGGCACAATGAAGTGGGTAA; reverse sequence (5′-3′): CTGAGCAAA GGCAATCAACA. AFP, forward sequence (5′-3′): GTACGGACATTCAGACTGCT; reverse sequence (5′-3′): TTGCTGCCTTTGTTTGGAAG. CK19, forward sequence (5′-3′): CAGCTTCTGAGACCAGGGTT; reverse sequence (5′-3′): GACTGGCGATAGCTGTAGGA. CDH2, forward sequence (5′-3′): CTCCATGTGCCGGATAGC; reverse sequence (5′-3′): TCTACAGACGCCTGAAGCAG. CLDN12, forward sequence (5′-3′): TTGAGCCCTCATCAAGCTCT; reverse sequence (5′-3′): CTCTCCCATGGCTGGATAAA. Caspase 3, forward sequence (5′-3′): TTCATTATTCAGGCCTGCCGAGG; reverse sequence (5′-3′): TTCTGACAGGCCATGTCATCCTCA. BCL-2, forward sequence (5′-3′): CATCAGGAAGGCTAGAGTTACC; reverse sequence (5′-3′): CAGACATTCGGAGACCACAC. Bax, forward sequence (5′-3′): -CCAGCTCTGAGCAGATCATG; reverse sequence (5′-3′): TGCTGGCAAAGTAGAAAAGG. β-actin, forward sequence (5′-3′): ACGCTCGGTCAGGATCTTC; reverse sequence (5′-3′): GAGACCTTCAACACCCC.

### 2.9. Western Blot Analysis

For Western blot analysis, after 48-h culture on different stiffness substrates and confluent growth cells were harvested. Briefly, total protein was obtained by radio immunoprecipitation assay (RIPA) lysis buffer. Protein quantification was performed by using bicinchoninic acid (BCA) protein assay kit (Beyotime, Beijing, China) to maintain the same load. Protein samples (40 μg) were electrophoretically separated by denaturing SDS-polyacrylamide gel (PAGE) and transferred to a polyvinylindene fluoride (PVDF) membrane. The membrane was blocked in 5% not-fat dry milk in TBST buffer (50 mM Tris-HCl, 100 mM NaCl, and 0.1% Tween-20, pH7.4), incubated with specific primary antibodies (F-actin, Abcam, Cambridge, UK; α-tubulin Abcam, Cambridge, UK; Albumin, Proteintech, Chicago, IL, USA) at 4 °C overnight, and washed with TBST buffer (10 min × 3). After that, the membrane was further incubated with corresponding horseradish peroxidase-conjugated second antibodies (second antibodies, all from Beyotime, Beijing, China). Densitometric analysis was performed using a VersaDoc imaging system with Quantity One software (Bio-Rad Laboratories, Hercules, CA, USA). GAPDH (Santa Cruz Biotechnology, Santa Cruz, CA, USA) was used as a loading control for total protein. All Western blots were run under the same experimental conditions and repeated at least three times to ensure reproducibility. For calculation of fold changes representing expression levels of protein under each treatment conditions, densitometry values of each band for the protein of interest was normalized with the loading control in each experiment.

### 2.10. Immunofluorescence Staining Analysis

Cells were cultured on different substrates stiffness; after 48 h, cells were rinsed in ice-cold PBS and fixed in 4% paraformaldehyde at room temperature for 20 min. After that, samples were permeabilized with 0.25% Triton-X-100 buffer at room temperature for 15 min. After blocking in 1% BSA for 1 h, cells were incubated with specific primary antibodies (F-actin, Abcam, UK; α-tubulin, Abcam, UK) at 4 °C overnight. Then, the samples were further incubated with fluorescent second antibody (Beyotime, Beijing, China) at 37 °C for 1-h and 5-min incubation with DAPI (Beyotime, Beijing, China) for nuclear staining in humidified box away from light. After washing with PBS, images were acquired using a Leica confocal microscope. Experiments were repeated at least three times.

### 2.11. Immunohistochemical Staining Analysis

For immunohistochemical (IHC) staining, sections were first washed by PBS (3 × 2 min). The sections were then fixed in 4% paraformaldehyde for 15 min and dried for 5 min at room temperature. Then, sections were incubated with 0.5% Triton X-100 for 20 min. After washing with PBS three times, sections were covered with 3% H_2_O_2_ for 15 min and then blocked with blocking reagent (QuickBlock™ Blocking Buffer, Beyotime, Beijing, China) for 1 h. Thereafter, the sections were incubated with specific primary antibody albumin (Albumin, Proteintech, Chicago, IL, USA) overnight at 4 °C. Sections were then incubated with second antibody (Beyotime, Beijing, China). Protein expression was detected using diaminobezidin (DAB) Detection Kit (PV9000, ZSGB-BIO, China). In negative controls, the primary antibody was replaced by isotype-matched IgG. The stained slides were mounted for observation and all images were taken using bright-field microscopy (Olympus-IX71, Tokyo, Japan).

### 2.12. Statistical Analysis

All experiments were repeated three times. Statistical analysis was performed by Origin software (Hampton, MA, USA). Statistical tests applied were One-Way analysis of variance (ANOVA) by Tukey-test. Data are presented as mean ± standard error of mean (SEM). *p*-values < 0.05 was deemed statistically significant.

## 3. Results

### 3.1. The Stiffness of PVA Hydrogels Could Be Modulated and Were Non-Cytotoxic

According to the demands of experiments, our hydrogels could be manufactured in a customized shape and using different patterns (Figure 1A). With AFM measuring, the Young’s modulus of the three hydrogels were 4.80 ± 0.11 kPa (soft), 21.27 ± 0.45 kPa (moderate), and 45.21 ± 0.16 kPa (stiff), which were representative of the tissue stiffness of a healthy liver or during the early stage or end-stage of liver fibrosis, respectively (Figure 1B). The rate of hepatocellular apoptosis cultured on the hydrogels was quantified by a Live/Dead assay. The results showed that minimal apoptosis was occurring on either of the three hydrogels. This implied that the PVA hydrogel was not cytotoxic to hepatocytes and was suitable for follow up experiments (Figure 1C).

### 3.2. The Concentration of Fibronectin Regulated the Trajectory and Displacement Distance of Hepatocytes

Considering that the concentration of ligands within the ECM can affect the migratory behavior of hepatocytes, we investigated the role of fibronectin on hepatocellular trajectory and displacement. PVA hydrogels of varying stiffness were coated with different concentrations of fibronectin (0.003 mg/mL, 0.005 mg/mL, 0.01 mg/mL, and 0.1 mg/mL). We found that the substrate stiffness and fibronectin concentrations had a significant effect on the migratory behavior of single hepatocytes. The migration trajectories of hepatocytes on soft, moderate, or stiff substrates coated with 0.003 mg/mL fibronectin were 90.89 μm, 86.38 μm, and 99.41 μm. The displacement distances were 24.34 μm, 21.64 μm, and 23.80 μm, respectively. The migration trajectories on 0.005 mg/mL fibronectin were 122.35 μm, 92.98 μm, and 67.98 μm. The displacement distances were 36.52 μm, 27.58 μm, and 19.95 μm, respectively. The migration trajectories on 0.01 mg/mL fibronectin were 699.32 μm, 565.77 μm, and 398.89 μm. The displacement distances were 219.52 μm, 154.96 μm, and 99.06 μm, respectively. However, 0.1 mg/mL fibronectin significantly inhibited hepatocyte migration, and the migration trajectories were 92.88 μm, 88.72 μm, and 81.60 μm. The displacement distances were 19.12 μm, 19.32 μm, and 21.22 μm, respectively (Figure 2A–C). Our results indicated that 0.01 mg/mL fibronectin promoted the optimal migratory behavior of hepatocytes; therefore, 0.01 mg/mL fibronectin was used in the following experiments.

### 3.3. Substrate Stiffness Regulated the Migration of Hepatocytes

As a typical epithelioid cell, the fused growth of hepatocytes is essential for tissue formation and reconstruction. However, the effect of increased stiffness in liver fibrosis on hepatocyte migration was unclear. Herein, the wound scratch assay was used to investigate the effect of substrate stiffness on the migratory behavior of fused growth hepatocytes. At the 6-h time point, the closure rates of the soft and moderate groups were significantly increased compared with the stiff group. There was no significant difference in hepatocellular migration between the three groups at 24 h. However, after 48 h and 72 h, the closure rates of the soft and moderate groups were significantly decreased compared to the stiff group (Figure 3A,B). These results indicated that, for fused cells, the migration of hepatocytes was regulated by the substrate stiffness and differed over time. Over a short period (6 h), the hepatocytes on the soft substrate displayed a faster closure rate, while, over a longer period (48 h and 72 h), the closure rate of the stiff group was significantly faster than the soft and moderate groups.

### 3.4. Substrate Stiffness Influenced Hepatocellular Migration by Increasing the Expression of F-Actin and α-Tubulin

In order to further explore the mechanisms by which substrate stiffness could influence hepatocellular migration, we measured the expression of cytoskeletal microfilament structural protein fibrous actin (F-actin) and tubulin. We found that the protein expression of F-actin significantly increased in the moderate group and stiff group compared to the soft group by 1.35-fold and 2.13-fold, respectively (Figure 4A,B). Immunofluorescence staining of F-actin showed consistent results (Figure 4C), and the fluorescence intensity of the moderate and stiff groups were significantly increased, which indicated that rigid substrate facilitated F-actin structure assembling.

Microtubules are an indispensable structural element of a cell which support and expand to maintain the overall shape of the cell. Alpha (α) and beta (β) tubulin dimers are the structural proteins of microtubules. There was significantly increased α-tubulin protein expression in the stiff group compared with the soft group (1.15-fold increase). There was no significant difference between the moderate group and the soft group (Figure 4D,E). Results of immunofluorescence staining for α-tubulin showed that the fluorescence density for α-tubulin in the stiff group was significantly increased (Figure 4F). These results suggested that the pathological stiffness observed in late liver fibrosis could significantly increase the expression of F-actin and α-tubulin in hepatocytes. This increase of skeletal proteins may play an important role in regulating cytoskeletal rearrangements and cell migration.

### 3.5. Substrate Stiffness Regulated the Function of Hepatocytes In Vitro

Previous studies have demonstrated that substrate stiffness could induce a contractile-to-synthetic phenotypic transition in vascular smooth muscle cells [19]. As a typical epithelioid cell, we speculated that substrate stiffness may affect hepatocellular function. Therefore, genes associated with the hepatocellular phenotype, epithelial-mesenchymal phenotype transformation, and apoptosis were examined. Our results showed that the expression of albumin, AFP, and CK19 were significantly decreased in the stiff group compared with the soft group (Figure 5). Protein expression and cell IHC of albumin further support the results (Figure 6). Additionally, the epithelial gene CLDN12 was down regulated and the mesenchymal gene CDH2 was up regulated compared with the soft group (Figure 5). The stiff substrate also induced hepatocyte apoptosis. Gene expression in the stiff group demonstrated that caspase 3 and Bax expression were increased and Bcl-2 expression was decreased compared with the soft group (Figure 5). These results indicated that the substrate stiffness mimicked the late stage of liver fibrosis could perturb hepatocellular function, facilitate epithelial-mesenchymal phenotype transformation, and induce apoptosis.

### 3.6. The Effect of 3D Substrate Stiffness on Hepatocyte Migration

To further explore the effect of substrate stiffness on hepatocyte migration in a 3D microenvironment, we used a stiffness controllable commercial Col-Tgel to mimic the stiffness of normal liver at the early and end stages of liver fibrosis. The hepatocytes were encapsulated and cultured for 48 h. Result of cell live/dead assay showed that the proportion of dead hepatocytes in the soft, moderate, and stiff groups were 2.68%, 11.61%, and 15.41%, respectively (Figure 7A,B). Results of F-actin staining showed that the cell morphology did not change significantly in the three groups, and they were spherical. Additionally, the expression of F-actin was decreased compared with that in the 2D substrate. Although the actin filaments are distributed in dots, however, in the 3D-stiff group, a small amount of filopodia was extended from the surface of the cell sphere (indicated by the yellow arrow). This result suggests that the 3D substrate stiffness was not significantly affected hepatocellular morphology, but the 3D-stiff group increased the expression of filopodia (Figure 7C). The migrator behavior of hepatocytes in different 3D substrate stiffness was also analyzed, and the result showed that 3D substrate stiffness significantly inhibited hepatocyte migration in three groups compared with the no gel group (control) (Figure 7D).

## 4. Discussion

Liver fibrosis is characterized by increased tissue stiffness. Liver cells aberrantly synthesize ECM proteins, such as type I and III collagens, proteoglycans, and non-collagenous glycoproteins, which in turn alter cellular phenotypes and perturb function [20]. At the end-stage, chronic liver fibrosis leads to cirrhosis and even deteriorates into hepatocellular carcinoma, which can be life-threatening [21]. Compelling research demonstrated that the status change of hepatic stellate cells from quiescent to activated is the initiator of liver fibrosis [22,23]. As the main cells in the liver, hepatocytes interact with all intrahepatic cell populations. However, the role of hepatocytes in liver fibrosis, especially how their fate is modulated by the rigidity of the ECM, is unknown. Therefore, to evaluate the effect of substrate stiffness on the migratory behavior and the function of hepatocytes, we established a stiffness-controllable PVA hydrogel platform in vitro which could mimic pathological liver tissue stiffness.

Our study showed that the PVA hydrogels of varying stiffness were biocompatible with the hepatocytes and non-cytotoxic, although the substrate stiffness significantly affected cellular morphology. Masha et al. demonstrated that human fibroblasts were elongated by the arrangement of focal adhesions on a stiff substrate. Furthermore, fibroblasts formed large and uniformly oriented focal adhesions, whereas only small and radially oriented adhesions were formed by cells cultured on the soft substrate [24]. Similarly, Wang et al. found that the epithelial cells pated on a 1.2 kPa substrate exhibited a more slender morphology compared to cells plated on a 90 Pa substrate [25]. As a typical epithelial cell, our results confirmed that there was an elongation and polarization change of the hepatocytes on a stiff substrate. Moreover, cells on a stiff substrate fostered larger amounts of lamellae compared with cells on a soft substrate. It is now appreciated that lamellae refers to two actin-rich subcellular structures, the filopodia and lamellipodia. They are assembled at the leading edge of the lamellae, which plays a key role in directional cellular migration [26]. For instance, filopodia are formed in fibroblasts to enhance cellular migration toward a substrate gradient [27], whereas the lamellipodia are vital for cell directional movement along an ECM ligand gradient [28]. Our results demonstrated that a stiff substrate significantly enhanced the expression of F-actin, the main component of filopodia and lamellipodia, on the leading edges of hepatocytes. It also increased the expression of α-tubulin, a substructure of the cell microtubules which helps maintain cell integrity. Liver fibrosis is a reversible wound-healing process, and cell migration is involved during the process. Our study demonstrated that hepatocytes cultured on a stiff substrate migrated faster compared to those cultured on a soft substrate, potentially due to the increased presence of lamellae structures in hepatocytes cultured on a stiff substrate.

Additionally, ligands, such as fibronectin and laminin, are essential for cells to sense the mechanical stimuli of the surrounding ECM and activate the relative migratory signaling pathways. Grasieli et al. demonstrated that fibronectin could promote cancerous oral squamous cell adhesion and migration. They found that fibronectin increased ras-related C3 botulinum toxin substrate 1, (Rac 1) activity and induced smaller adhesions, resulting in fast single cell migration in both 2D and 3D microenvironments [29]. With the increasing stiffness of the ECM in liver fibrosis, the concentration of fibronectin also increased. Therefore, we detected the effect of different concentrations of fibronectin on hepatocyte migration. Our results indicated that high concentrations of fibronectin promoted hepatocellular migration trajectory and displacement on a stiff substrate.

It is well known that cellular function, homeostasis, and even stem cell differentiation contributes to ECM stiffness [23,30,31]. Huang et al. demonstrated that varying substrate stiffness regulated the cellular uptake of nanoparticles [32]. Another study showed that substrate stiffness regulated the differentiation of endothelial progenitor cellular arterial-venous cells by activating Ras/Mek (rat sarcoma/MAPK/ERK kinase) pathway [33]. A recent study of human liver sinusoidal endothelial cells (hLSECs) indicated that fenestrations and CD31, important features of the hLSECs phenotype, were completely lost on the stiff substrate, indicating that liver fibrosis may perturb the function of resident cells [34]. Our gene expression analysis indicated that, with increasing substrate stiffness, hepatocytes exhibited an epithelial to mesenchymal transformation. Expression of epithelial genes, such as CLDN12, was down-regulated, while the expression of a mesenchymal gene, CDH2, was significantly up-regulated. Moreover, the expressions of cell apoptotic genes were also elevated in stiff substrate. Albumin is one of the classical protein markers of hepatocellular function, and our results showed that the expression of albumin was significantly inhibited on a stiff substrate. Taken together, our study indicated that substrate stiffness was an important regulator of hepatocellular function. An increase in substrate stiffness altered the phenotype of hepatocytes and enhanced cellular apoptosis. Furthermore, a stiff substrate attenuated the function of hepatocytes. To further explore the effect of substrate stiffness on hepatocytes migration, in this study, we also used a stiffness controllable 3D hydrogel. Our results showed that hepatocytes had a higher proportion of cell death and expression of filopodia, which further confirmed the results of 2D substrate. However, 3D substrate stiffness inhibited the migration of hepatocytes in the three 3D groups compared with the control group. The possible reason may be due to the structure of our 3D hydrogel.

However, there are some limitations to this study. First, although our results revealed that hepatocytes cultured on a stiff substrate stiffness were more motile following the promotion of F-actin and α-tubulin rich structures, the underlying mechanisms of this phenomenon are unknown. Further studies may focus on the signaling pathways employed by hepatocytes in response to ECM mechanical stimuli. Second, this study only explored the effects of substrate stiffness on hepatocytes; however, hepatocytes can interact with all intrahepatic cell populations in liver. Therefore, we may want to explore the communication between hepatocytes and intrahepatic cell populations in the liver on a stiff substrate.

## 5. Conclusions

In summary, we established a stiffness-controllable PVA-hydrogel system in vitro which mimics the changes in physical tissue stiffness during hepatic fibrosis. We also focused on the often-overlooked hepatocytes. We demonstrated that a stiff substrate facilitated hepatocellular migration, attenuated the phenotype and function of hepatocytes, and enhanced apoptosis. Taken together, these results may provide a new insight into investigating the mechanism of ECM stiffness in liver fibrosis.

## Figures and Tables

**Figure 1 polymers-12-01903-f001:**
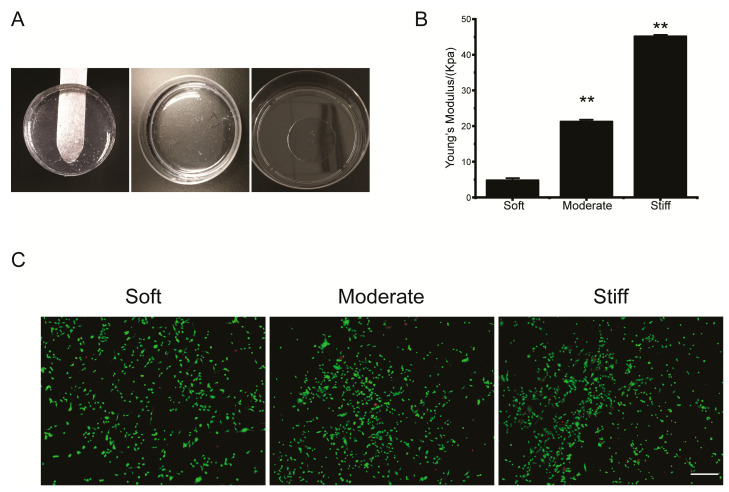
Polyvinyl alcohol (PVA) hydrogels showed stiffness controllable mechanical properties and were not cytotoxic. (**A**) PVA hydrogels were fabricated in different molds. (**B**) Stiffness of PVA hydrogels were tested by atomic force microscopy (AFM). Data were presented as the mean ± SEM (*n* = 30). ** represents *p* < 0.01. (**C**) Live/dead assay for hepatocytes cultured on soft, moderate, and stiff substrates. Representative fluorescence images of live/dead cell staining. Calcein-AM (green) indicates live cells, while PI (red) indicates dead cells. Scale bar = 40 μm.

**Figure 2 polymers-12-01903-f002:**
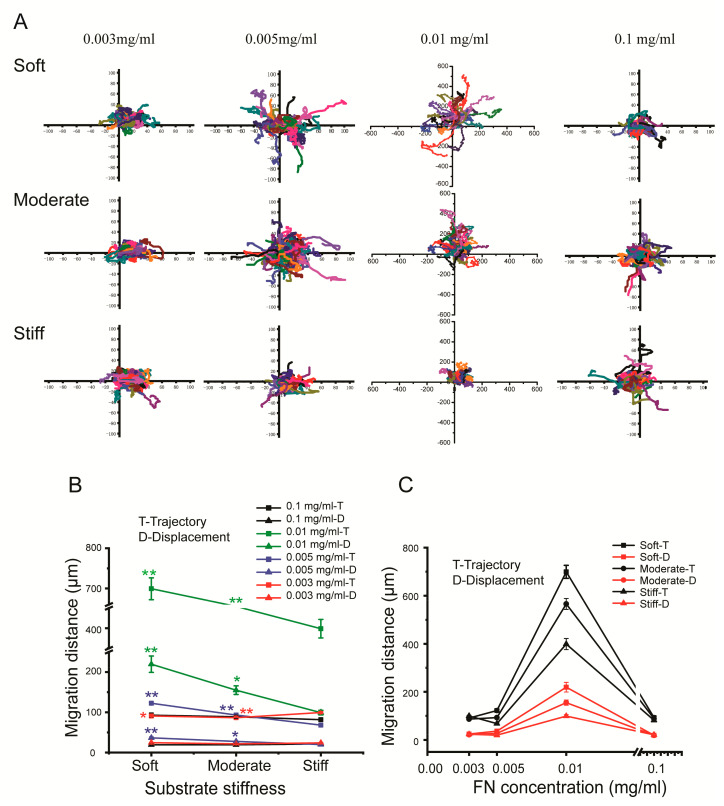
Cellular migration trajectory of hepatocytes cultured on substrates of different stiffness and different concentrations of fibronectin. (**A**) Cellular migration trajectory of hepatocytes cultured on substrates of different stiffness and varying concentrations of fibronectin, (*n* ≥ 30). Cellular migration trajectory and displacement are plotted based on increasing substrate stiffness (**B**) and fibronectin concentration (**C**). Soft and moderate groups compared to the stiff group. Data were presented as the mean ± SEM (*n* ≥ 30). * represents *p* < 0.05; ** represents *p* < 0.01.

**Figure 3 polymers-12-01903-f003:**
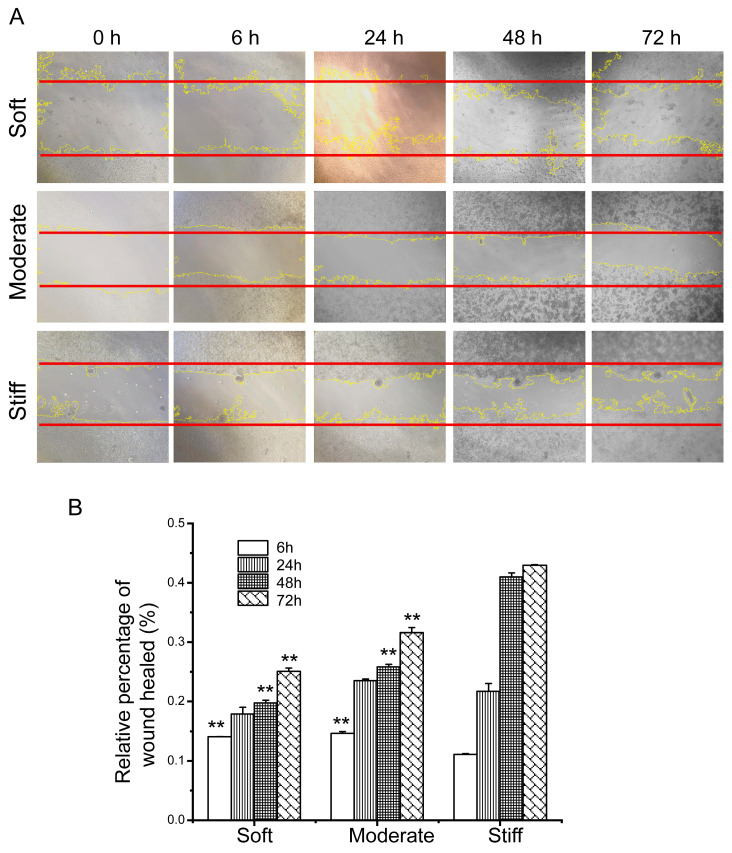
Substrate stiffness affects cell migration of confluent hepatocytes. (**A**) Representative scratch assay images of hepatocytes on soft, moderate, and stiff substrates at different time points at 40× objective. (**B**) Relative percentage of the wound healed. The red line represents the initial scratch distance, and the yellow line is the cell boundary line when ImageJ automatically analyzes the scratches. Soft and moderate groups compared to the stiff group. Data presented as the mean ± SEM (*n* = 3). ** represents *p* < 0.01.

**Figure 4 polymers-12-01903-f004:**
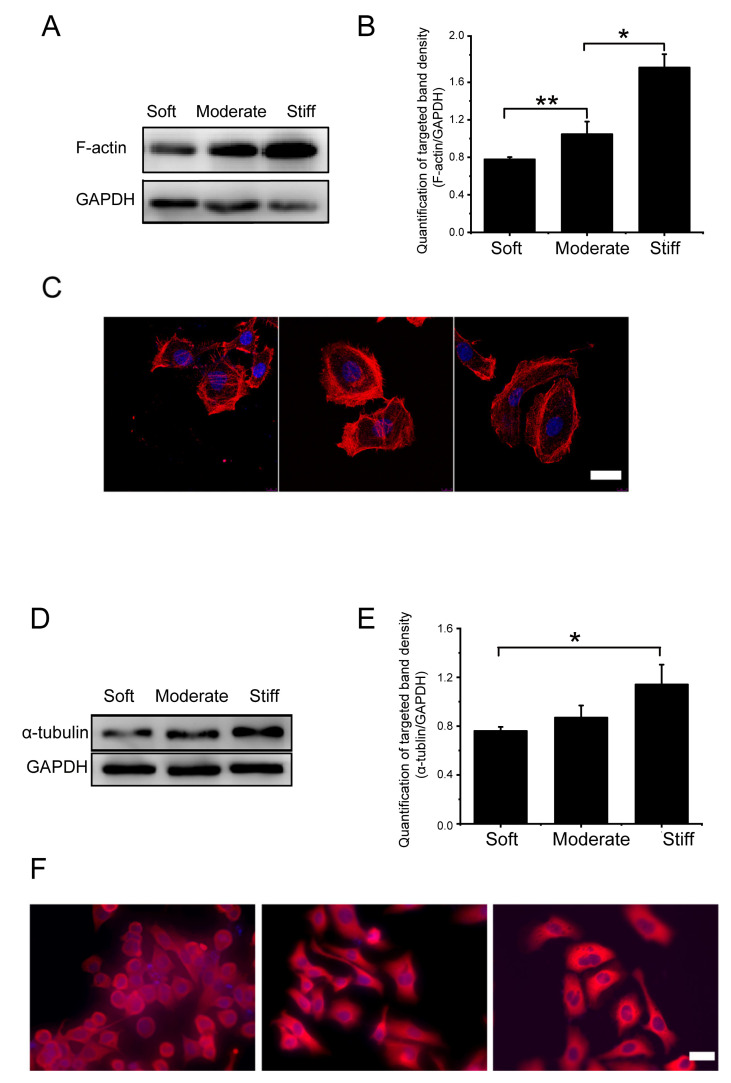
Substrate stiffness affects F-actin and α-tubulin expression of hepatocytes cultured on soft, moderate, and stiff substrates for 48 h. (**A**) Protein synthesis of F-actin was detected by Western blot. (**B**) Band densities were analyzed by Quantity One software. (**C**) Protein synthesis of F-actin was detected by immunofluorescence staining; scale bar = 20 μm. (**D**) Protein synthesis of α-tubulin was detected by Western blot. (**E**) Band density was analyzed by Quantity One software. (**F**) Protein synthesis of α-tubulin was detected by immunofluorescence staining; scale bar = 40 μm. Moderate and stiff groups compared to the soft group. Data presented as the mean ± SEM (*n* = 3). * represents *p* < 0.05; ** represents *p* < 0.01.

**Figure 5 polymers-12-01903-f005:**
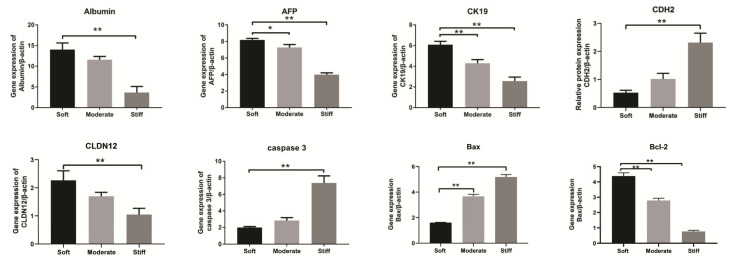
Gene expression was analyzed by RT-qPCR. Data are presented as the mean ± SEM (*n* = 3). Moderate and stiff groups compared to the soft group. * represents *p* < 0.05; ** represents *p* < 0.01.

**Figure 6 polymers-12-01903-f006:**
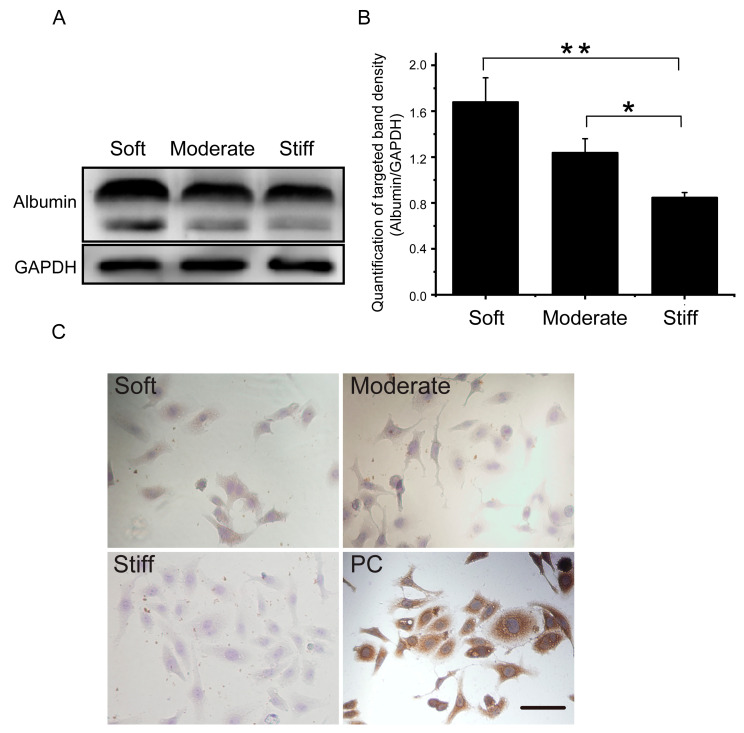
Substrate stiffness affects albumin expression in hepatocytes cultured on soft, moderate, and stiff substrates for 48 h. (**A**) Protein synthesis of albumin was detected by Western blot. (**B**) Band density was analyzed by Quantity One software. (**C**) Protein synthesis of albumin was detected by immunocytochemistry. Cells cultured on a plastic culture dish were used as a positive control; scale bar = 25 μm. Soft and moderate groups compared to the stiff group. Data were presented as the mean ± SEM (*n* = 3). * represents *p* < 0.05; ** represents *p* < 0.01.

**Figure 7 polymers-12-01903-f007:**
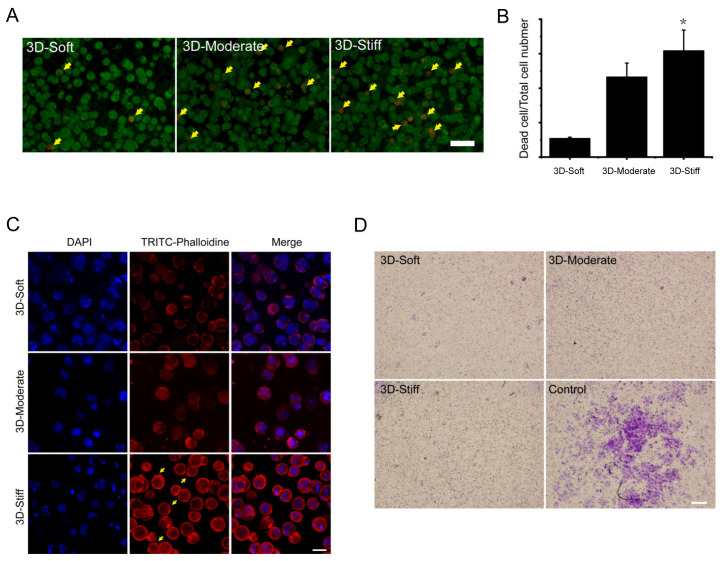
Live/dead assay for hepatocytes cultured on 3D soft, moderate and stiff substrates for 48 h. (**A**) Fluorescence images of live/dead cells staining. Calcein-AM (green) indicates live cells, while PI (red) indicates dead cells, bar = 40 μm. (**B**) Statistics of dead cell ratio (dead cell/ total cell). Compared to the soft group. Data presented as the mean ± SEM (*n* = 3). * represents *p* < 0.05. (**C**) Cytoskeleton image of hepatocytes cultured on 3D soft, moderate, and stiff substrates. Cells were stained for TRITC (tetramethyl rhodamine isothiocynate)-phalloidine (red) and cell nucleus (blue), bar = 25 μm. (**D**) Images of the migration of hepatocytes grown in different stiffness 3D gel, no gel group as control, bar = 200 μm, *n* = 3.

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
