# Peer review of "The Effect of Matrix Stiffness on Human Hepatocyte Migration and Function—An In Vitro Research"

_polymers, 2020, doi:10.3390/polym12091903_

Round 1
Reviewer 1 Report
The manuscript must be revised; the authors should describe the aims and the potential applications. English needs to be revised.
There are numerous of grammatically incorrect sentences that need to be rewrited, just to list a few:
In page 1, Abstract line 13 and 14, 'To investigate the effect of different ECM stiffness on hepatocytes function. We designed a biomimetic...' , Unclear sentence. Please rewrite it and make it clear by correcting the sentence so that will be grammatical correct.
In page 1, Abstract line 14 and 15 "...using a stiffness-controllable polyvinyl alcohol (PVA) hydrogel, which coated with..." Unclear sentence. Please rewrite it and make it clear by correcting the sentence so that will be grammatical correct.
In page 1, Abstract line 16 and 17 "...were selected for corresponding to the health liver tissue, the early and late stages of fibrotic liver tissues." Unclear sentence. Please rewrite it and make it clear by correcting the sentence so that will be grammatical correct.
In page 1, Abstract line 18, "Our result showed that the hydrogel showed well biocompatible to hepatocytes survival." Please correct the sentence.
in page 7, Conclusion line 305-307 "...in vitro which mimicking the physical tissue stiffness changes..." Please rewrite the whole sentence, grammatically incorrect.
Page 8, Figure 1C: The fluorescent micrographs of life/dead assay for hepatocytes are very unclear - you can not see green or red colored cells. Please correct and sharpen the images (adjust contrast/brightness)...
Page 9, Figure 2: In the Figure caption you forgot to mark C. "(B and C) Cellular migration 332 trajectory and displacement are plotted based on increasing substrate stiffness and based on 333 fibronectin concentration."
Overall, introduction provides sufficient background, research design is appropriate and methods are adequately described. Results are clearly presented and conclusions are supported by the results.
Author Response
Reviewer 1
Question1. The manuscript must be revised; the authors should describe the aims and the potential applications. English needs to be revised.
Answer: Thank you for your kindly advice. We have added some contents about the aims and the potential applications of our work:
Hepatocytes are the main parenchymal cells and functional units of the liver. Liver fibrosis is a complex process. Previous studies have shown that hepatocytes contributed to the process of liver fibrosis through changing migratory behavior and biological functions. Hepatocytes can undergo epithelial-mesenchymal phenotypic transition, which increase the migratory ability and changes in the mechanical properties of hepatocytes can regulate cell migration and mechanical signal transduction [14-17]. However, at present, whether matrix stiffness, a factor that significantly changes in the process of liver fibrosis, can regulate hepatocytes participate in the process of liver fibrosis is unknown. Previous study has demonstrated that hepatocytes cultured on soft heparin gels were secreting 5-fold higher levels of albumin compared to cells cultured on stiff heparin [18]. However, research concerning the effects of pathological ECM stiffness on hepatocytes is still underdeveloped.
- Baimakhanov Z, Sakai Y, Yamanouchi K, Hidaka M, Soyama A, Takatsuki M, et al. Spontaneous hepatocyte migration towards an endothelial cell tube network. Journal of tissue engineering and regenerative medicine. 2018;12:e1767-e71.
- Pellicoro A, Ramachandran P, Iredale JP, Fallowfield JA. Liver fibrosis and repair: immune regulation of wound healing in a solid organ. Nature reviews Immunology. 2014;14:181-94.
- Guicciardi ME, Malhi H, Mott JL, Gores GJ. Apoptosis and Necrosis in the Liver. Compr Physiol. 2013;3:977-1010.
- Mattei G, Magliaro C, Giusti S, Ramachandran SD, Heinz S, Braspenning J, et al. On the adhesion-cohesion balance and oxygen consumption characteristics of liver organoids. PloS one. 2017;12:e0173206.
- You, J., et al., Characterizing the effects of heparin gel stiffness on function of primary hepatocytes. Tissue Eng Part A, 2013. 19(23-24): p. 2655-63.
You can see the modified part in the manuscript at line 52-63 red font.
Additionally, we also find a professional language company to revise this manuscript.
Question2. There are numerous of grammatically incorrect sentences that need to be rewrited, just to list a few:
In page 1, Abstract line 13 and 14, 'To investigate the effect of different ECM stiffness on hepatocytes function. We designed a biomimetic...' , Unclear sentence. Please rewrite it and make it clear by correcting the sentence so that will be grammatical correct.
In page 1, Abstract line 14 and 15 "...using a stiffness-controllable polyvinyl alcohol (PVA) hydrogel, which coated with..." Unclear sentence. Please rewrite it and make it clear by correcting the sentence so that will be grammatical correct.
In page 1, Abstract line 16 and 17 "...were selected for corresponding to the health liver tissue, the early and late stages of fibrotic liver tissues." Unclear sentence. Please rewrite it and make it clear by correcting the sentence so that will be grammatical correct.
In page 1, Abstract line 18, "Our result showed that the hydrogel showed well biocompatible to hepatocytes survival." Please correct the sentence.
Answer: Thank you for your kindly advice. We have rewrited this sentence as “To investigate the effect of ECM stiffness on hepatocyte migration and function. We designed an easy fabricated polyvinyl alcohol (PVA) hydrogel which its stiffness can be controlled by changing the concentration of glutaraldehyde. Three stiffness of hydrogels corresponding to the health of liver tissue, early stage and end stage of fibrosis were selectd. These were 4.8 kPa (soft), 21 kPa (moderate) and 45 kPa (stiff). For hepatocytes attachment, the hydrogel was coated with fibronectin. To evaluate the optimal concentration of fibronectin, hydrogel was coated with 0.1 mg/mL, 0.01 mg/mL, 0.005 mg/mL, or 0.003 mg/mL fibronectin and the migratory behavior of single hepatocyte cultured on different concentrations of fibronectin was analyzed. Our result confirmed the hydrogel biocompatibility with high hepatocytes survival.”
You can see the modified part in the part of abstract at line 11-20 red font.
Question 3: In page 7, Conclusion line 305-307 "...in vitro which mimicking the physical tissue stiffness changes..." Please rewrite the whole sentence, grammatically incorrect.
Answer: Thank you for your suggestion. We have rewrited the sentence as” In summary, we establish a stiffness-controllable PVA-hydrogel system in vitro which mimics the changes in physical tissue stiffness during hepatic fibrosis.”
You can see the revised sentence in the part of conclusion at line 316-317 red font.
Question 4: Page 8, Figure 1C: The fluorescent micrographs of life/dead assay for hepatocytes are very unclear - you can not see green or red colored cells. Please correct and sharpen the images (adjust contrast/brightness)...
Answer: We are sorry for the problems of our pictures. We have re-adjusted the size and quality of all the pictures in our article.
Question 5: Page 9, Figure 2: In the Figure caption you forgot to mark C. "(B and C) Cellular migration 332 trajectory and displacement are plotted based on increasing substrate stiffness and based on 333 fibronectin concentration."
Answer: Thank you for pointing out our mistake. We have marked Figure C in the figure legend. You can see the revised part in the manuscript at line 345-347 red font.

Reviewer 2 Report
The manuscript entitled " Designing a stiffness controllable hydrogel to investigate the effect of substrate stiffness on human hepatocyte migration and function" describes the design of 2D model based on polyvinyl alcohol with tunable mechanical stiffness to assess the effect of matrix properties on hepatocyte migration. The mechanical stiffness of the surface was controlled by changing the concentration of glutaraldehyde. The experimental design for the hypothesis mentioned in the introduction as well as the results gained by the experiments are relatively appropriate. However, there is major concern about the hypothesis of the project along with some minor comments as below which need to be considered:
Major comment:
- This work has been done on a 2D model. Why 2D? Is there any work on the effect matrix properties on 3D which is a more relevant model to in vivo? There are many 3D matrixes developed and available with tunable properties (including stiffness) as a 3D model for in vitro studies.
Minor comments:
- Title: The title does not seem the best for this paper. For example, the word “stiffness” has been repeated two times in a short title. That would be nice to come up with better title.
- L18: better to change the sentence structure. For example: “Our result confirmed the hydrogel biocompatibility with high hepatocytes survival”
- L44-45: The unmet need mentioned in this sentence is not clear. Please be more specific about the previous studies on hepatocytes migration and what the unmet needs you are addressing in this study.
- L49: Change the sentence to “affects the function of hepatocyte and will trigger”
- L67: change “solved” to “dissolved”
- Figure 1. The quality of the figure is poor and out of size proportion (Fig 4 and 2 are stretched)
- All figures need to be revised. There some problem with either size proportion or quality of all figures. Last figure caption is “Figure 1”. Should not it be “Figure 6”
- L176: after “behavior of hepatocytes” insert comma “,” not dot “.”
- L296-300: The sentences grammatical structure needs to be corrected.
- L306: Change “mimicking” to “mimics”

Author Response
Reviewer 2
Major comment
Question 1: This work has been done on a 2D model. Why 2D? Is there any work on the effect matrix properties on 3D which is a more relevant model to in vivo? There are many 3D matrixes development and available with tunable properties (including stiffness) as a 3D model for in vitro studies.
Answer: Thank you for your kindly advice. In this study, we used a tunable stiffness of 2D model mimicking different stages of liver fibrosis to explore the effects of liver stiffness on hepatocyte migration and biological functions. The reasons for choosing a 2D model are that the 2D plane can better track the migratory trajectory of a single hepatocyte and more convenient for cell imaging; the 2D model can purely investigate the effect of matrix stiffness on hepatocyte migration. The current established in vitro 3D models of liver fibrosis include 3D spheroids, 3D organoids, 3D bioprinting, Bioreactors and organ-on-chip, 3D vascularized models, decellularized tissue matrices and precision cut tissue slices. The research purposes of these models are focused on exploring cell-cell or cell-ECM interactions, mimicking the architecture and microenvironment of natural or fibrotic liver tissue, or screening drugs (Table 1.). At present, there are few reports on the use of 3D models with tunable stiffness to explore the effects of matrix stiffness on liver cell migration and biological functions.
Table 1. Uses of current classes of liver fibrotic tissue models
|
Model |
Uses |
Refs |
|
2D culture |
A. Screening of antifibrotic agent and lineage-mapping studies B. Cell migration and mechanobiology studies C. Coculture studies to investigate paracrine interactions |
[1-3] |
|
3D spheroids |
A. Drug screening and toxicity studies B. Mechanistic studies understanding the in vivo and patient mimic fibrotic microenvironment |
[4, 5] |
|
3D organoids |
A. Adult stem cells (ASCs) derived organoids: disease modeling. B. iPSC-derived organoids: fibrosis development modeling C. Drug screening and toxicity studies |
[6, 7] |
|
3D bioprinting |
A. Fabrication of 3D biomimetic scaffold similar to in vivo situation B. Replication and study of induced fibrogenesis mechanisms at the cellular and physiological level C. 3D cell–cell and cell–ECM interaction studies |
[8-10] |
|
Bioreactors and organ-on-chip |
A. Studies of induced fibrogenesis mechanisms at the cellular and physiological level under dynamic environment B. Evaluating effects of drugs, due to dynamic tissue microenvironment |
[11, 12] |
|
3D vascularized models |
A. In vitro study of vascular remodeling and cell migration B. Drug screening |
[13] |
|
Decellularized tissue matrices |
A. In vitro culture substrate mimicking native or fibrotic disease microenvironment |
[14, 15] |
|
Precision cut tissue slices |
A. In vitro culture substrate mimicking native or fibrotic disease microenvironment |
[16] |
Additionally, we are currently conducting relevant 3D experiments. We encapsulate hepatocytes in 3D hydrogels with different stiffness. We already obtained some results and we are still working on the 3D part work. But this part of work was not ready for publish yet. Part of our 3D work were showed below:
① The 3D-Stiff group significantly up-regulates the proportion of dead hepatocytes.
Figure 1 Live/dead assay for hepatocytes cultured on 3D soft, moderate and stiff substrates for 48 h. (A) Fluorescence images of live/dead cells staining. Calcein-AM (green) indicates live cells while PI (red) indicates dead cells, bar = 40 μm. (B) Statistics of dead cell ratio (dead cell/ total cell). Compared to the soft group. Datas presented as the mean ± SEM (n= 3). * represents P < 0.05; ** represents P < 0.01.
② The 3D-Stiff group up-regulates the expression of hepatic cytoskeleton microfilaments, which can enhance cell force signal transduction.
Figure 2 Cytoskeleton image of hepatocytes cultured on 3D soft, moderate and stiff substrates. Cells were stained for TRITC-phalloidine (red) and cell nucleus (blue), bar = 25 μm.
③ There is no significant change in the migration of hepatocytes in different matrix stiffness, which may be due to the structure of our 3D hydrogel.
Figure 3 Images of the migration of hepatocytes grown in different stiffness 3D gel, no gel group (PC) as control, bar = 200 μm, n=3.
- De Minicis, S. et al. (2007) Gene expression profiles during hepatic stellate
cell activation in culture and in vivo. Gastroenterology 132, 1937–1946
- Seki, E. et al. (2007) TLR4 enhances TGF-β signaling and hepatic fibrosis.
Nat. Med. 13, 1324
- Sawitza, I. et al. (2009) The niche of stellate cells within rat liver.
Hepatology 50, 1617–1624
- Bell, C.C. et al. (2016) Characterization of primary human hepatocyte
spheroids as a model system for drug-induced liver injury, liver function and
disease. Sci. Rep. 6, 25187
- Leite, S.B. et al. (2016) Novel human hepatic organoid model enables
testing of drug-induced liver fibrosis in vitro. Biomaterials 78, 1–10
- D.G. Nguyen, J. Funk, J.B. Robbins, C. Crogan-Grundy, S.C. Presnell, T.
Singer, A.B. Roth, Bioprinted 3D primary liver tissues allow assessment of
organ-level response to clinical drug induced toxicity in vitro, PLoS One 11
(2016) e0158674.
- L.M. Norona, D.G. Nguyen, D.A. Gerber, S.C. Presnell, E.L. LeCluyse,
Editor's high- light: modeling compound-induced fibrogenesis in vitro using
three-dimensional bioprinted human liver tissues, Toxicol. Sci. 154 (2016)
354–367.
- Norona, L.M. et al. (2016) Editor’s highlight: modeling compound- induced
fibrogenesis in vitro using three-dimensional bioprinted human liver tissues.
Toxicol. Sci. 154, 354–367
- Nguyen, D.G. et al. (2016) Bioprinted 3D primary liver tissues allow
assessment of organ-level response to clinical drug in- duced toxicity in
vitro. PLoS One 11, e0158674-e0158674
- Ma, X. et al. (2016) Deterministically patterned biomimetic human
iPSC-derived hepatic model via rapid 3D bioprinting. Proc. Natl. Acad. Sci.
113, 2206
- Illa, X. et al. (2014) A novel modular bioreactor to in vitro study
the hepatic sinusoid. PLoS One 9, e111864-e111864
- Jeong, G.S. et al. (2016) Viscoelastic lithography for fabricating
self-organizing soft micro-honeycomb structures with ultra-high
aspect ratios. Nat. Commun. 7, 11269
- Jin, Y. et al. (2018) Vascularized liver organoids generated using
induced hepatic tissue and dynamic liver-specific microenvironment as a
drug testing platform. Adv. Funct. Mater. 28, 1801954 Minor comment
- B.E. Uygun, A. Soto-Gutierrez, H. Yagi, M.L. Izamis, M.A. Guzzardi, C.
Shulman, J. Milwid, N. Kobayashi, A. Tilles, F. Berthiaume, M. Hertl, Y.
Nahmias, M.L. Yarmush, K. Uygun, Organ reengineering through
development of a transplantable recellularized liver graft using
decellularized liver matrix, Nat. Med. 16 (2010) 814-U120.
- G. Mazza, K. Rombouts, A. Rennie Hall, L. Urbani, T. Vinh Luong, W.
Al-Akkad, L. Longato, D. Brown, P. Maghsoudlou, A.P. Dhillon, B. Fuller, B.
Davidson, K. Moore, D. Dhar, P. De Coppi, M. Malago, M. Pinzani,
Decellularized human liver as a natural 3D-scaffold for liver bioengineering
and transplantation,
- Paish, H.L. et al. (2019) A bioreactor technology for modeling fibrosis in
human and rodent precision-cut liver slices. Hepatology 70, 1377–1391
Question 2. Title: The title does not seem the best for this paper. For example, the word “stiffness” has been repeated two times in a short title. That would be nice to come up with better title.
Answer: Thank you for your advice. We changed the title as “The effect of matrix stiffness on human hepatocyte migration and function—an in vitro research”.
You can see the modified part in the part of title at line 2-3 red font.
Question 3. L18: better to change the sentence structure. For example: “Our result confirmed the hydrogel biocompatibility with high hepatocytes survival”.
Answer: We are very grateful for the suggestion you made for us. We have rewritten this sentence as “Our result confirmed the hydrogel biocompatibility with high hepatocytes survival”.
You can see the modified part in the abstract at line 19-20 red font.
Question 4. L44-45: The unmet need mentioned in this sentence is not clear. Please be more specific about the previous studies on hepatocytes migration and what the unmet needs you are addressing in this study.
Answer: Thank you for telling us the inadequacies of this article. We have added some contents in the part of introduction.
For instance, human skin fibroblasts grow on polyethylene glycol (PEG) gel, when the stiffness of PEG gel is from 95 Pa to 4.3 kPa, the cell migration speed decreases from 0.81 μm/min to 0.38 μm/min [10]. As one of the features of liver fibrosis, the stiffness of liver tissue is gradually increased during the progress of fibrosis. However, very few studies focus on the effect of matrix stiffness on the migratory behavior of hepatocytes during liver fibrosis.
Hepatocytes are the main parenchymal cells and functional units of the liver. Liver fibrosis is a complex process, which affects hepatocyte function and will trigger a series of events [11-13]. Previous studies have shown that hepatocytes contributed to the process of liver fibrosis through changing migratory behavior and biological functions. Hepatocytes can undergo epithelial-mesenchymal phenotypic transition, which increase the migratory ability and changes in the mechanical properties of hepatocytes can regulate cell migration and mechanical signal transduction [14-17]. However, at present, whether matrix stiffness, a factor that significantly changes in the process of liver fibrosis, can regulate hepatocytes participate in the process of liver fibrosis is unknown. Previous study has demonstrated that hepatocytes cultured on soft heparin gels were secreting 5-fold higher levels of albumin compared to cells cultured on stiff heparin [18]. However, research concerning the effects of pathological ECM stiffness on hepatocytes is still underdeveloped.
- Ghosh K, Pan Z, Guan E, Ge SR, Liu YJ, Nakamura T, et al. Cell adaptation to a physiologically relevant ECM mimic with different viscoelastic properties. Biomaterials. 2007;28:671-9.
- Yang, L., et al., Transforming growth factor-beta signaling in hepatocytes promotes hepatic fibrosis and carcinogenesis in mice with hepatocyte-specific deletion of TAK1. Gastroenterology, 2013. 144(5): p. 1042-1054 e4.
- Das, N., et al., Melatonin protects against lipid-induced mitochondrial dysfunction in hepatocytes and inhibits stellate cell activation during hepatic fibrosis in mice. J Pineal Res, 2017. 62(4).
- Zhang, X.W., et al., Antagonism of Interleukin-17A ameliorates experimental hepatic fibrosis by restoring the IL-10/STAT3-suppressed autophagy in hepatocytes. Oncotarget, 2017. 8(6): p. 9922-9934.
- Baimakhanov Z, Sakai Y, Yamanouchi K, Hidaka M, Soyama A, Takatsuki M, et al. Spontaneous hepatocyte migration towards an endothelial cell tube network. Journal of tissue engineering and regenerative medicine. 2018;12:e1767-e71.
- Pellicoro A, Ramachandran P, Iredale JP, Fallowfield JA. Liver fibrosis and repair: immune regulation of wound healing in a solid organ. Nature reviews Immunology. 2014;14:181-94.
- Guicciardi ME, Malhi H, Mott JL, Gores GJ. Apoptosis and Necrosis in the Liver. Compr Physiol. 2013;3:977-1010.
- Mattei G, Magliaro C, Giusti S, Ramachandran SD, Heinz S, Braspenning J, et al. On the adhesion-cohesion balance and oxygen consumption characteristics of liver organoids. PloS one. 2017;12:e0173206.
- You, J., et al., Characterizing the effects of heparin gel stiffness on function of primary hepatocytes. Tissue Eng Part A, 2013. 19(23-24): p. 2655-63.
You can also see the modified part in the manuscript at line 46-63 red font.
Question 5. L49: Change the sentence to “affects the function of hepatocyte and will trigger”
Answer: Thank you for your correction. We have revised this sentence as “Liver fibrosis is a complex process, which affects hepatocyte function and will trigger a series of events”.
You can see the revised sentence in the manuscript at line 52-53 red font.
Question 6. L67: change “solved” to “dissolved”
Answer: Many thanks for your correction. We have changed this word. You can see it in the part of 2.1. PVA hydrogel preparation at line 75 red font.
Question 7. Figure 1. The quality of the figure is poor and out of size proportion (Fig 4 and 2 are stretched)
Answer: Thank you for your advice. We have re-adjusted the size and quality of all the pictures in our article.
Question 8. All figures need to be revised. There are some problems with either size proportion or quality of all figures. Last figure caption is “Figure 1”. Should not it be “Figure 6”.
Answer: We are sorry for the problems of our pictures. We have re-adjusted the size and quality of all the pictures in our article. We revised the Figure 1 to Figure 6 at line 242 and 369 red font.
Question 9. L176: after “behavior of hepatocytes” insert comma “,” not dot “.”
Answer: Thank you for your correction. We have replaced the dot with a comma. You can see it at line 186 -188 red font.
Question 10. L296-300: The sentences grammatical structure needs to be corrected.
Answer: Thank you for your suggestion. We have rewritten these sentences as
Our gene expression analysis indicated that with increasing substrate stiffness, hepatocytes exhibited an epithelial to mesenchymal transformation. Expression of epithelial genes such as CLDN12 was down-regulated while the expression of a mesenchymal gene, CDH2, was significantly up-regulated. Moreover, the expressions of cell apoptotic genes were also elevated in stiff substrate. Albumin is one of the classical protein markers of hepatocellular function, and our results showed that the expression of albumin was significantly inhibited on a stiff substrate. Taken together, our study indicated that substrate stiffness was an important regulator of hepatocellular function. An increase in substrate stiffness altered the phenotype of hepatocytes and enhanced cellular apoptosis. Furthermore, a stiff substrate attenuated the function of hepatocytes.
However, there are some limitations to this study. First, although our results revealed that hepatocytes cultured on a stiff substrate stiffness were more motile following the promotion of F-actin and α-tubulin rich structures, the underlying mechanisms of this phenomenon are unknown. Further studies may focus on the signaling pathways employed by hepatocytes in response to ECM mechanical stimuli.
You can also see the revised sentence in the part of result at line 298-311 red font.
Question 11. L306: Change “mimicking” to “mimics”
Answer: Many thanks for your correction. We have changed this word. You can see it in the part of conclusion at line 316 red font.

Round 2
Reviewer 2 Report
The revised manuscript looks much better and the authors effort for modifications is greatly appreciated. This is also great to see that the 3D model is being considered for this project. Therefore, that would be nice to combine some of the 3D model results with the current 2D ones to have a more interesting and comprehensive publication for others.
Author Response
Comments and Suggestions for Authors
The revised manuscript looks much better and the authors effort for modifications is greatly appreciated. This is also great to see that the 3D model is being considered for this project. Therefore, that would be nice to combine some of the 3D model results with the current 2D ones to have a more interesting and comprehensive publication for others.
Answer We do appreciate your kindly suggestion. We have added some contents and results about the 3D model.
- To further explore the effect of substrate stiffness on hepatocyte migration, we used a stiffness controllable commercial 3D Col-Tgel which has similar substrate stiffness like PVA hydrogel. (line 19-21, red font)
- Hepatocytes in stiff 3D hydrogel showed a higher proportion of cell death and expression of filopodia. (line 27-28, red font)
- Additionally, the migrator behavior of hepatocytes in 3D substrate was also analyzed. (line 72-73, red font)
- 4. Cell culture on the 3D hydrogel
A stiffness controllable 3D Col-Tgel (Weihui Biotechnology, China) was used for analyzing the migrator behavior of hepatocytes in 3D substrate stiffness. Preheated the Collagen-I (Col) solution at 55°C for about 5 minutes until the collagen gel melted into a liquid state, and then cool it down at room temperature for 5 minutes before proceeding. The TG powder was taken out from -80°C and warmed at room temperature, and then dissolved in 100 μl sterile deionized water according to the instructions. After removing the medium, mixed the digested hepatocytes and the Col solution by gently pipetting to avoid air bubbles. Added the TG solution according to the proportion and mixed gently by pipetting. In this study, 20 μl of 3D Col-Tgel mixture was seeded with 2×103 hepatocytes. Dropped 20 μl of the 3D Col-Tgel-cell mixture dropwise into the 24-well plate cell special circular slide (diameter 14 mm), and incubate in a cell incubator at 37°C for 45 min. After the gel solidified, placed it into the 24-well plate and then added 1 mL of 10% FBS medium into the plate to ensure that the medium completely covered the gel, and cultured cells for subsequent testing. (line 104-116, red font)
- and 3D hydrogel (line 118, red font)
- 7. Cell migration analysis in 3D hydrogel
The assay was performed using 24-well transwell chambers containing polycarbonate filters with a pore size of 8 μm. The lower compartment was filled with 10% FBS medium. 1×104 hepatocytes were seeded in 100 μl 3D gels on the upper compartment of the chamber. After incubation at 37°C for 96 h, the cells and gels on the upper surface of the filter were removed by wiping. Cells that passed the filter were fixed with 4% paraformaldehyde for 20 min, stained with 0.5% crystal violet for 15 min. Images were captured by using phase contrast microscopy (Olympus-IX71, Japan). (line 128-135, red font)
- 6 The effect of 3D substrate stiffness on hepatocyte migration
To further explore the effect of substrate stiffness on hepatocyte migration in a 3D microenvironment, we used a stiffness controllable commercial Col-Tgel to mimic the stiffness of normal liver, the early and end stages of liver fibrosis. The hepatocytes were encapsulated and cultured for 48 hours. Result of cell live/dead assay showed that the proportion of dead hepatocytes in the soft, moderate and stiff groups were 2.68%, 11.61% and 15.41% respectively (Fig.7 A and B). Result of F-actin staining showed that the cell morphology did not change significantly in the three groups, and they were spherical. Additionally, the expression of F-actin was decreased compared with that in the 2D substrate. Although the actin filaments are distributed in dots, however, in the 3D-stiff group, a small amount of filopodia was extended from the surface of the cell sphere (indicated by the yellow arrow). This result suggests that the 3D substrate stiffness was not significantly affected hepatocellular morphology, but the 3D-stiff group increased the expression of filopodia (Fig. 7 C). The migrator behavior of hepatocytes in different 3D substrate stiffness was also analyzed and the result showed that 3D substrate stiffness significantly inhibited hepatocyte migration in three groups compared with the no gel group (control) (Fig. 7 D). (line 273-287, red font)
Figure 7 Live/dead assay for hepatocytes cultured on 3D soft, moderate and stiff substrates for 48 h. (A) Fluorescence images of live/dead cells staining. Calcein-AM (green) indicates live cells while PI (red) indicates dead cells, bar = 40 μm. (B) Statistics of dead cell ratio (dead cell/ total cell). Compared to the soft group. Datas presented as the mean ± SEM (n= 3). * represents P < 0.05. (C) Cytoskeleton image of hepatocytes cultured on 3D soft, moderate and stiff substrates. Cells were stained for TRITC-phalloidine (red) and cell nucleus (blue), bar = 25 μm. (D) Images of the migration of hepatocytes grown in different stiffness 3D gel, no gel group as control, bar = 200 μm, n=3. (line 419-427, red font)
- To further explore the effect of substrate stiffness on hepatocytes migration, in this study we also used a stiffness controllable 3D hydrogel. Our results showed that hepatocytes had a higher proportion of cell death and expression of filopodia, which further confirmed the results of 2D substrate. However, 3D substrate stiffness inhibited the migration of hepatocytes in the three 3D groups compared with the control group. The possible reason may be due to the structure of our 3D hydrogel. (line 345-351, red font)
